# A Novel Method for Early Fatigue Damage Diagnosis in 316L Stainless Steel Formed by Selective Laser Melting Technology

**DOI:** 10.3390/ma16093363

**Published:** 2023-04-25

**Authors:** Xiaoling Yan, Xiujian Tang

**Affiliations:** 1College of Artificial Intelligence, Beijing Technology and Business University, Beijing 102488, China; 2Science and Technology on Remanufacturing Laboratory, Armored Forces Academy of PLA, Beijing 100072, China; tangxiujian@sina.com

**Keywords:** nonlinear ultrasonic, early fatigue damage, selective laser melting, signal processing, reliability

## Abstract

Early fatigue damage is an important factor affecting the service safety of 316L stainless steel parts formed by selective laser melting (SLM) technology. Nonlinear ultrasonic testing for early fatigue damage in SLM 316L stainless steel specimens was carried out. A new method for evaluation of early fatigue damage based on nonlinear ultrasonic testing was proposed. Empirical mode decomposition (EMD) was applied to the unsteady ultrasonic testing signal, and the signal was decomposed into multiple intrinsic mode functions (IMFs) that meet certain conditions; then, the specific IMF (ESI) containing the effective fatigue damage information was extracted. Lastly, fast Fourier transform (FFT) was applied to the specific IMF signal to obtain the required information to evaluate the damage in the measured part caused by fatigue. The results of nonlinear ultrasonic testing agreed well with transmission electron microscope experimental analysis and theoretical model of acoustic nonlinearity caused by dislocations. The change in nonlinear ultrasonic testing results reflected the generation and evolution of dislocation structure during the low-cycle fatigue regime of the SLM 316L stainless steel specimen and revealed the early fatigue damage mechanism of this metal part. Compared with the classical FFT method, the EMD-ESI-FFT method is more sensitive in identifying the early damage in SLM 316L stainless parts induced by fatigue loading, which is equivalent to improving the early fatigue damage identification and diagnosis ability and can better ensure the service safety of important metal parts.

## 1. Introduction

Selective laser melting (SLM) [1,2] is a promising rapid prototyping technology that can be used to fabricate any complex structural part with a molding density of nearly 100%. In the past decade, its application has expanded from the original conceptual mold design to the aerospace, biomedical, and automotive fields. It is very important to ensure the structural integrity and service safety of components manufactured using SLM technology. Therefore, early damage to SLM components should be detected before significant defects occur. Micro defects caused by fatigue damage are prone to expansion, leading to the destruction of the entire component. Therefore, nondestructive testing of early fatigue damage is crucial.

Traditional NDT techniques, such as X-ray [3,4], ultrasonic [5,6,7], eddy current [8,9], penetrant [10], and magnetic powder [11] techniques, excel in the detection of macroscopic defects, but their effect in the detection of micro defects caused by early fatigue damage proves to be dissatisfactory. Under the action of fatigue load, the micro defects produced in the early stage of fatigue very easily expand into macro defects, thus leading to the damage of the entire component. Weak changes in degradation and evolution of material properties or microcracks can cause nonlinear effects of ultrasonic propagation in the medium, which resulted in the development of nonlinear ultrasonic testing techniques [12,13]. Nonlinear ultrasonic testing techniques can effectively evaluate the degree of material performance degradation or parameters such as the size and state of microcracks and provide a new means for early detection and prevention of material damage. The ultrasonic signal collected by the receiving transducer contains the nonlinear response generated by micro defects. The specific information of defects can be extracted through signal processing to achieve the purpose of detection and evaluation of material damage and functional degradation. Because the collected signals usually have nonlinear and unsteady characteristics [14,15], including many interference signals, eliminating interference signals and extracting effective information characterizing material damage are key techniques for nonlinear ultrasonic testing. 

The nonlinear ultrasonic testing methods used in practice mainly include higher harmonic [16,17], beam mixing and modulation [18,19,20], resonant frequency spectrum analysis, and nonlinear ultrasonic phased array imaging methods [21]. The higher harmonic detection method is more widely used because of its simple and convenient technology. Classical nonlinear acoustics [22,23] shows that the second harmonics are mainly induced by the harmonicity of the crystal lattice of materials. The nonclassical nonlinearity effect is caused by micro defects (cracks, pores) in metallic material. Relevant studies [24,25,26,27,28,29,30,31] have shown that fatigue damage of metal materials (dislocations, microcracks, etc.) is strongly correlated with nonlinear ultrasonic higher harmonic signals. Using higher harmonic detection technology to evaluate early fatigue damage of metal materials is a research hotspot in material evaluation. 

When the higher harmonic testing method is used for the detection of early fatigue damage in a metallic material, the signal processing method used is usually fast Fourier transform (FFT). FFT has high accuracy in analyzing steady-state signals, and the signal can be transformed from the time domain to the frequency domain; thus, the information required to evaluate the damage degree in the measured part caused by fatigue can be obtained [32]. A deep learning model and higher-order spectral analysis were applied to the ultrasonic data collected from fatigue experiments [33]. However, the ultrasonic signal received is unstable. FFT signal processing technology cannot effectively filter the interference signals in the ultrasonic signal, and the generalization and interpretability of deep learning are not strong. In view of the above limitations, a new signal processing method (EMD-ESI-FFT) is proposed in this paper; the signal is decomposed into multiple IMFs that meet certain conditions, and the specific IMF containing fatigue damage information is extracted according to the method described in this paper. Then, FFT is applied to the specific IMF signal to extract the required fatigue damage information. The results of higher harmonic testing agree well with transmission electron microscope experimental analysis and theoretical model of acoustic nonlinearity caused by dislocations. The change in nonlinear ultrasonic testing results reflects the generation and evolution of dislocation structure during the low-cycle fatigue regime of the SLM 316L stainless steel specimen and reveals the early fatigue damage mechanism of this metal part. Compared with the classical FFT method, EMD-ESI-FFT method is more sensitive in identifying the early damage in SLM 316L stainless parts induced by fatigue loading, which is equivalent to improving the early fatigue damage identification and diagnosis ability and can better ensure the service safety of important metal parts.

## 2. Theoretical Model of Acoustic Nonlinearity Caused by Early Fatigue Damage

The early fatigue damage of metal materials mainly comes from the dislocation in the crystal. With the further deepening of the fatigue degree, the dislocation density continues to increase, and the resulting nonlinear effect also tends to be obvious. As shown in Figure 1, the mechanism of its nonlinear generation can be explained by the dislocation model [34,35,36]. For the dislocation chord MIN with M and N as nodes, *r* is the radius of curvature, 2*θ* is the center angle of the arcuate dislocation chord, the dislocation length is 2*L*, and the dislocation chord MIN moves in the slip plane under the action of stress, and its shear strain is
(1)$$\gamma_{d}=\frac{\Lambda b}{2L}S$$
where *r_d_* represents the strain caused by dislocation, Λ is the dislocation density, *b* is Burger’s vector, and *S* is the area swept by the dislocation when moving in the slip plane (shaded part in Figure 1). The expression of *S* is
(2)$$S=r^{2}(\theta -\frac{1}{2}\sin(2\theta ))$$

Shear stress *τ* is
(3)$$\tau =\frac{T}{rb}=\frac{Gb}{2r}$$
where *T* is the tension of the dislocation string, *G* is the shear modulus, and sin(2θ) can be expanded in series about *θ* ≈ *L*/*r*. The stress–strain relationship caused by dislocation can be obtained as
(4)$$\gamma_{b}=\frac{T}{rb}=\frac{\Lambda L^{2}}{3G}\cdot \tau +\frac{\Lambda L^{4}}{15G^{3}b^{2}}\cdot \tau^{3}+\ldots$$

According to *ε_d_* = Ω *γ_d_*, *τ = Rσ*, Ω is the conversion coefficient of shear strain *γ_d_* and normal strain *ε_d_*, and *R* is the conversion coefficient of shear stress *τ* and normal stress *σ*, from which we can obtain that the normal strain caused by dislocation is
(5)$$\varepsilon_{1}=\frac{2\Lambda L^{2}}{3G}\cdot \sigma +\frac{8\Lambda L^{4}\Omega R^{3}}{15G^{3}b^{2}}\cdot \sigma^{3}+\ldots$$
because the normal strain of the lattice inside the solid is
(6)$$\varepsilon_{2}=\frac{1}{E_{1}}\cdot \sigma +\frac{E_{2}}{E_{1}}\cdot \sigma^{2}+\ldots$$
where *E*_1_ and *E*_2_ are second-order and third-order elastic constants, respectively. The normal strain in the solid is the sum of the internal dislocation and the normal strain caused by the lattice; it can be expressed as
(7)$$\varepsilon_{3}=(\frac{2\Lambda L^{2}}{3G}+\frac{1}{E_{1}})\sigma +\frac{E_{2}}{E_{1}}\cdot \sigma^{2}+\frac{8\Lambda L^{4}\Omega R^{3}}{15G^{3}b^{2}}\cdot \sigma^{3}+\ldots$$

The occurrence of dislocation not only increases the linear term of the stress–strain relationship within the solid, but also increases the higher-order nonlinear term. With the increase in dislocation density Λ, the coefficient of the nonlinear term becomes larger, and the resulting acoustic nonlinear phenomenon is more obvious. We have verified this through experiments. The experimental results are shown in Section 4.3.

## 3. Experimental Procedures

### 3.1. Material and Specimen Preparation

The specimens were prepared by selective laser melting 3D printing technology. The raw material was powdered 316L stainless steel; the scanning electron microscope (SEM) photograph of the powder is shown in Figure 2. The particle size of the spherical powder was 8–50 μm; the apparent density of the powder was 4.43 g/cm^3^. Table 1 shows the chemical composition of powdered 316L stainless steel.

The AM400 additive manufacturing system (Renishaw plc, Gloucestershire, UK) was used to prepare the specimens. Table 2 shows the main fabrication parameters. The building orientation is vertical. Four specimens (C1 to C4) were prepared (shown in Figure 3a). Figure 3b shows the size of the specimen (size in millimeters). The maximum loading stress in high-cycle fatigue tests should be less than the yield strength of the measured specimen. To obtain the yield strength of the measured specimen, tensile tests were carried out on specimens C1 to C3; then, the intermittent low cycle fatigue test and nonlinear ultrasonic test for specimen C4 were performed, and its early fatigue damage was evaluated.

### 3.2. Experimental Setup and Data Collection Method

The results of tension tests showed that the yield strength of the measured specimen was 534 MPa, so 430 MPa was set as the maximum loading stress. The force applied in the fatigue test was tensile load, and the waveform was a sine wave curve. The stress ratio was set to 0.1, and the loading frequency was 10 Hz. An Instron 8801 (Instron, Boston, MA, USA) experimental instrument was used in the tension test. 

The experimental setup for nonlinear ultrasonic detection is shown in Figure 4. RAM-5000 SNAP (RITEC Inc., Warwick, RI, USA) is highly integrated; a computer, matching resistance, transducers, and an oscilloscope work well with RITEC RAM-5000 SNAP.

The nonlinear ultrasonic test for specimen C4 was performed before the tension–tension fatigue test. Firstly, specimens were polished with waterproof abrasive paper of 200 to 2000 meshes, and the surfaces of transducers and measured specimens were wiped. Then, the couplant was applied to the contact surfaces of transducers and measured specimen. The couplant used in this experiment was a universal coupling agent (Zhongxi, Beijing, China). As shown in Figure 4, the transmitting and receiving transducers were fixed separately on the proper opposite surface of the measured specimen. Then, an elastic band was used to fix the transducers and the measured specimen together, to avoid the installation positions being changed because of other operations. An ultrasonic transducer with a 5 MHz center frequency was used for excitation. A receiving transducer with a 10 MHz center frequency was used at the receiving end. According to the testing method described above, fatigue tests were conducted for specimen C4, when the number of fatigue loading was 1000, 2000, 3000, 4000, 5000, 6000, 7000, 8000, and 9000, the fatigue loading test was suspended and the fatigue load was kept unchanged, and nonlinear ultrasonic tests were conducted for specimen C4. In order to reduce the accidental error in the experiment, the receiving signals were acquired five times for the same fatigue cycles.

## 4. Method for Fatigue Damage Diagnosis in 316L Stainless Specimen

In the process of nonlinear ultrasonic testing, the ultrasonic signal collected by the receiving transducer contains the nonlinear response generated by the fatigue damage. The specific fatigue damage information can be extracted through signal processing; because the collected signals usually have nonlinear and unsteady characteristics, including many interference signals, eliminating interference signals and extracting effective information characterizing material damage are key techniques for nonlinear ultrasonic testing.

### 4.1. Fast Fourier Transform (FFT)

When the higher harmonic testing method is used for the detection of fatigue damage in metal materials, the signal processing method used is usually FFT. FFT is applied to the aperiodic continuous time signal *x*(*t*), and the continuous spectrum *x*(ω) can be obtained [37]:(8)$$x\left(\omega \right)=\int_{-\infty }^{\infty }x\left(t\right)e^{-j\omega t}dt$$

The nonlinear ultrasonic testing signal is the discrete signal *x*(*n*) (*n* = 0, 1,2, …, *N* − 1), and *x*(*n*) is the discrete sampling value of the continuous signal *x*(*t*). The spectrum of *x*(*n*) can be obtained by discrete Fourier transform (DFT); the principle of DFT is as follows:(9)$$x\left(k\right)=\overset{N-1}{\underset{n=0}{\sum }}x\left(n\right)W_{n}^{kn}$$
where *k* = 0, 1, 2, …, *N* − 1, $W_{n}=e^{-j\frac{2\pi }{N}}$.

FFT has high accuracy in analyzing steady-state signals. The signal can be transformed from the time domain to the frequency-domain to obtain the required fatigue damage information. Figure 5 shows the received time-domain signal and the Fourier spectra for specimen C4 subjected to 2000 and 8000 cycles.

It can be seen from Figure 5b,d that the fundamental wave (the center frequency is 5 MHz and the maximum amplitude is expressed in *A*_1_) and second harmonic (the center frequency is 10 MHz and the maximum amplitude is expressed in *A*_2_) can be obtained in the Fourier spectrum of the received signal. The first-order perturbation solution [38] of the nonlinear acoustic wave equation shows that the nonlinear coefficient *β* = *A*_2_/*A*_1_^2^, and *β* can be calculated according to the Fourier spectrum of the received signal. In the process of detection for early fatigue damage in the measured parts, the ultrasonic signal received is unstable. FFT signal processing technology cannot effectively filter the interference signals in the ultrasonic signal. Therefore, FFT has limitations in the signal processing of nonlinear ultrasonic testing. In view of the above limitations, a new signal processing method (EMD-ESI-FFT) is proposed in this paper.

### 4.2. EMD-ESI-FFT

When EMD-ESI-FFT signal processing method is used, firstly, the nonlinear ultrasonic testing signal is decomposed into multiple IMFs that meet certain conditions, and fatigue damage information only exists in a specific IMF; then, the specific IMF (ESI) containing the fatigue damage information is extracted according to the method described in this paper; lastly, FFT is applied to the specific IMF signal to obtain fatigue damage information.

#### 4.2.1. Empirical Mode Decomposition (EMD)

EMD is the process of transforming unsteady signals into steady signals. An unsteady signal can be regarded as a combination of multiple IMFs [39]. Extracting the instantaneous frequency of a single IMF can characterize the characteristics of each frequency band in the unsteady signal. An IMF needs to meet the following requirements:Throughout the data segment, the result of subtracting the number of extreme points and zero crossings of the signal is less than or equal to 1.At any point on the time axis, the mean value of the upper and lower envelopes fitted by the maximum and minimum points is zero. That is, the upper and lower envelopes are locally symmetrical with the time axis as the axis of symmetry.

EMD of the original signal *x*(*n*) is carried out based on the above conditions. The specific decomposition process is as follows:The maximum points in *x*(*n*) are screened out, and cubic spline interpolation is performed on them to fit the upper envelope of *x_max_*(*n*). Similarly, the lower envelope *x_min_*(*n*) of *x*(*n*) can be obtained. The mean value of the upper and lower envelope is obtained at each time point:
(10)$$m\left(n\right)=\frac{x_{min}\left(n\right)+x_{max}\left(n\right)}{2}$$

2.Subtract the amplitude of *m*(*n*) from the original signal *x*(*n*) to obtain a new signal *h*(*n*):
*h*(*n*) = *x*(*n*) − *m*(*n*)(11)

3.If *h*(*n*) does not meet the IMF requirements, *x*(*n*) = *h*(*n*), and continue the steps (1) and (2). When *h*(*n*) meets the requirements, *c*_1_(*n*) = *h*(*n*), and extract *c*_1_(*n*) from *x*(*n*):
*r*_1_(*n*) = *x*(*n*) − *c*_1_(*n*)(12)

4.Repeat steps (1) and (2) for signal *r*_1_(*n*) to obtain *c*_2_(*n*). By analogy, continue to repeat steps (1) and (2) to obtain all the components that meet the IMF requirements.


(13)
$$\left.\begin{array}{c} r_{1}\left(n\right)-c_{2}\left(n\right)=r_{2}\left(n\right)\\ \vdots \\ r_{L-1}\left(n\right)-c_{L}\left(n\right)=r_{L}\left(n\right) \end{array}\right\}\;$$


5.When the residual component *r_L_*(*n*) becomes a monotonic function sequence or a constant series, the loop ends and the EMD is completed. The original signal *x*(*n*) is decomposed into a series of steady-state signals *c*_1_(*n*), *c*_2_(*n*), *c*_3_(*n*), …, *c_L_*(*n*) and a residual component *r_L_*(*n*); the residual component *r_L_*(*n*) represents the overall change trend of the signal.


(14)
$$x\left(n\right)=\overset{L}{\underset{i=1}{\sum }}c_{i}\left(n\right)+r_{L}\left(n\right)$$


In the process of signal processing, it is not convenient or feasible to judge whether *h*(*n*) meets the IMF requirements. Therefore, Huang et al. [33] proposed the standard deviation *SD* to judge whether a loop ends.
(15)$$SD=\overset{N-1}{\underset{n=0}{\sum }}\left[\frac{{\left|h_{\left(k-1\right)}\left(n\right)-h_{k}\left(n\right)\right|}^{2}}{h_{\left(k-1\right)}^{2}\left(n\right)}\right]$$
where *h*_*k*−1_(*n*) and *h_k_*(*n*) are two adjacent screening results. In order to meet the IMF requirements for linearity and stability, the value of *SD* is generally between 0.2 and 0.3.

In the process of nonlinear ultrasonic testing, the nonlinear effect caused by fatigue damage has the characteristics of localization, and the distortion process time is relatively short. FFT cannot fully characterize the process. EMD can filter out the redundant interference signal in the original signal, which is more conducive to analyzing the nonlinear distortion process and evaluating the damage degree of materials. Figure 6 shows the results of EMD for the time-domain signals.

#### 4.2.2. Extract the Specific IMF and Obtain the Fatigue Damage Information

The principle of early damage detection based on higher harmonic testing is shown in Figure 7. During the detection process, the transmitting transducer excites a fixed-frequency ultrasonic wave, which is called the fundamental wave. The fundamental wave is incident in the tested specimen and interacts with the micro defects caused by fatigue damage to generate higher harmonic signals, which are received by the receiving transducer. The received ultrasonic signals have nonlinear and unsteady characteristics; EMD is the process of transforming unsteady signals into steady signals. The unsteady signal can be regarded as a combination of multiple IMFs. A single IMF can characterize the characteristics of each frequency band in the unsteady signal. Due to fatigue damage information being reflected in high-order harmonic signals, if *A*_2_ (the amplitude of second harmonic) is maximal in one IMF, the second harmonic of this IMF can most effectively reflect the fatigue damage information. The method to extract the specific IMF is as follows:*c_k_*(*n*) *= argmax*(*A*_2_(*c_i_*(*n*)))(*i* = 1, 2, 3… *L*)(16)
where *c_k_*(*n*) (*n* = 0, 1, 2, …, *N* − 1) is the specific IMF, *c_i_*(*n*) (*i* =1, 2, 3, …, *L*) are the intrinsic mode functions of the original signal, and *A*_2_(*c_i_*(*n*)) is the amplitude of the second harmonic in *c_i_*(*n*).

In Figure 6a, IMF1 and IMF4~IMF7 are obviously interference signals, so fatigue damage information may exist in IMF2 and IMF3. Therefore, FFT transformations are applied to IMF2~IMF3, and Figure 8 shows the corresponding frequency-domain analysis results; the analysis results show that the fundamental wave (center frequency is 5 MHz) can be seen in IMF2 and IMF3, but the center frequency of fundamental wave has shifted in IMF2, and the second harmonic (center frequency is 10 MHz) exists in IMF3. There is no second harmonic in IMF2, so the fatigue damage information exists in IMF3. In fact, according to formula (16), IMF3 is the specific IMF containing the fatigue damage information, so the method given in this study is correct.

In Figure 6b, according to the waveform and amplitude of each IMF, IMF3~IMF8 are obviously interference signals, so fatigue damage information may exist in IMF1 and IMF2. FFT transformations are applied to IMF1~IMF2, and Figure 9 shows the corresponding frequency-domain analysis results; the analysis results reveal that the fundamental wave (center frequency is 5 MHz) can be seen in IMF1 and IMF2, but the center frequency of the fundamental wave has shifted in IMF2, and the second harmonic (center frequency is 10 MHz) exists in IMF1 and IMF2, but the second harmonic is extremely weak in IMF2, so IMF1 contains the fatigue damage information. In fact, according to formula (16), IMF1 is the specific IMF containing the fatigue damage information, so the method given in this study is reliable.

### 4.3. Results and Discussion

#### 4.3.1. Comparison of FFT and EMD-ESI-FFT

The ultrasonic testing signals were processed by FFT and EMD-ESI-FFT methods. Figure 10 shows the analysis results. Compared with FFT, the EMD-ESI-FFT method was adopted for the measured specimen with the same fatigue damage; the fundamental wave amplitude *A*_1_ is reduced, the second harmonic amplitude *A*_2_ is significantly larger, and the wave crest of the second harmonic is clearer, which makes the value of nonlinear coefficient *β* = *A*_2_/*A*_1_^2^ (representing the fatigue damage degree) larger. Thus, the sensitivity of the EMD-ESI-FFT method for early fatigue damage evaluation of the SLM 316L stainless part is verified. This is of great significance for improving the sensitivity of early fatigue damage evaluation of SLM 316L stainless parts, which can effectively avoid major accidents caused by fatigue damage.

Figure 11 shows the received time-domain signal and EMD-ESI-FFT result of the time-domain signal for specimen C4 not subjected to fatigue loading. The second harmonic is exhibited in the EMD-ESI-FFT result. The couplant and experimental setup are nonlinear in the experiment, which is the reasonable explanation for the emergence of nonlinear acoustic effect *β*_0_ in the measured specimen before the tension–tension fatigue test. To improve the reliability of the higher harmonic detection method, the nonlinear effect *β*_0_ caused by the couplant and experimental setup should be considered; the normalized nonlinear coefficient is used to evaluate the early fatigue damage in the measured specimen in this study. The ultrasonic testing results shown in Figure 12 indicate that *β*/*β*_0_ increases significantly with the accumulation of fatigue cycles. Obviously, it can be concluded that the enhancement of *β*/*β*_0_ is a result of the accumulation of damage in the measured specimen caused by fatigue cycles, and compared with FFT, the EMD-ESI-FFT method is more sensitive in identifying the early fatigue damage in the SLM 316L stainless part.

#### 4.3.2. Transmission Electron Microscope (TEM) Experimental Results

The experimental results show that as the number of fatigue cycles increased from 0 to 9000, the *β*/*β*_0_ increased accordingly, and compared with FFT, the EMD-ESI-FFT method is more sensitive in identifying the early fatigue damage in the SLM 316L stainless part. The theoretical model of acoustic nonlinearity caused by early fatigue damage indicates that early fatigue damage of metal materials mainly comes from dislocations [38,39,40] in the crystal. Owing to the couplant and experimental setup being nonlinear in the experiment, to better interpret the enhancement of *β*/*β*_0_ resulting from the accumulation of the dislocations, the microstructure of the measured specimen was analyzed using a transmission electron microscope (TEM). 

The TEM photographs of the tested specimen subjected to different fatigue cycles are shown in Figure 13. As shown in Figure 13a, when the fatigue cycle number reached 1000, the plane dislocation with monopole distribution appeared in the microstructure of the tested specimen. Figure 12 shows the change trend of *β*/*β*_0_ during low-cycle fatigue loading. When the fatigue cycle number changed from 0 to 1000, *β*/*β*_0_ continued to increase. When the fatigue cycle number increased, fatigue damage gradually accumulated, which was mainly reflected in the gradual increase in dislocation density in the microstructure of the fatigue specimen. Figure 13b,c indicate that the density of plane dislocation increased significantly, and dislocation entanglement gradually appeared. The fatigue cycles were within 1000~7000, *β*/*β*_0_ still maintained a rapid growth trend. When the number of fatigue cycles was 7000~9000, the increasing trend of *β*/*β*_0_ slowed down and the overall trend was still increasing. At this stage, dislocation veins and dislocation walls dominated by dipole distribution appeared in the microstructure of the specimen, as shown in Figure 13d,e. According to the theoretical model of acoustic nonlinearity caused by early fatigue damage, with the increase in dislocation density, the resulting acoustic nonlinear phenomenon is more obvious [40,41,42,43]. The results of nonlinear ultrasonic testing agree well with TEM experimental analysis and theoretical model of acoustic nonlinearity caused by dislocations. Therefore, the change in *β*/*β*_0_ reflects the generation and evolution of dislocation structure during the low-cycle fatigue regime of the SLM 316L stainless steel specimen and reveals the early fatigue damage mechanism of this metal part. Eliminating interference signals and extracting effective information characterizing material damage are key techniques for nonlinear ultrasonic testing. Compared with the classical FFT method, the EMD-ESI-FFT method proposed in this paper has higher sensitivity for the detection of early fatigue damage, which is equivalent to improving the early fatigue micro-damage identification and diagnosis ability and can better ensure the service safety of important metal parts.

## 5. Conclusions

Eliminating interference signals and extracting effective information characterizing material damage are key techniques for nonlinear ultrasonic testing. The EMD-ESI-FFT method for fatigue damage diagnosis was proposed in this paper. In the process of nonlinear ultrasonic testing, the nonlinear acoustic effect caused by fatigue damage has the characteristics of localization, and the distortion process time is relatively short. EMD-ESI-FFT can filter out the redundant interference signals in the original signal, which is more conducive to analyzing the nonlinear distortion process and evaluating the damage degree of materials.According to the principle of higher harmonic generation, the fatigue damage information is usually extracted by the classical FFT method. The ultrasonic signals have nonlinear and unsteady characteristics; FFT signal processing technology cannot effectively filter the interference signals in the ultrasonic signal. Theoretical analysis reveals that EMD is the process of transforming unsteady signals into steady signals. The unsteady signal can be regarded as a combination of multiple IMFs. Extracting the instantaneous frequency of a single IMF can characterize the characteristics of each frequency band in the unsteady signal due to fatigue damage information only existing in a specific IMF. The method to extract the specific IMF is described in this paper. Experimental results indicate that compared with FFT, the EMD-ESI-FFT method is more sensitive in identifying the early damage in SLM 316L stainless parts induced by fatigue loading, which is equivalent to improving the early fatigue micro-damage identification and diagnosis ability and can better ensure the service safety of important metal parts.According to the theoretical model of acoustic nonlinearity caused by early fatigue damage, the results of nonlinear ultrasonic testing agree well with TEM experimental analysis and the theoretical model of acoustic nonlinearity caused by dislocations. Therefore, the change in *β*/*β*_0_ reflects the generation and evolution of dislocation structure during the low-cycle fatigue regime of the SLM 316L stainless steel specimen and reveals the early fatigue damage mechanism of this metal part.The future development direction is the combination of nonlinear ultrasonic testing technology with big data and cloud computing to realize the full-time structural health monitoring of important components. Through the fusion analysis and data mining of historical data, the remaining life of components and materials can be predicted and the safety and reliability of important components in service can be improved.

## Figures and Tables

**Figure 1 materials-16-03363-f001:**
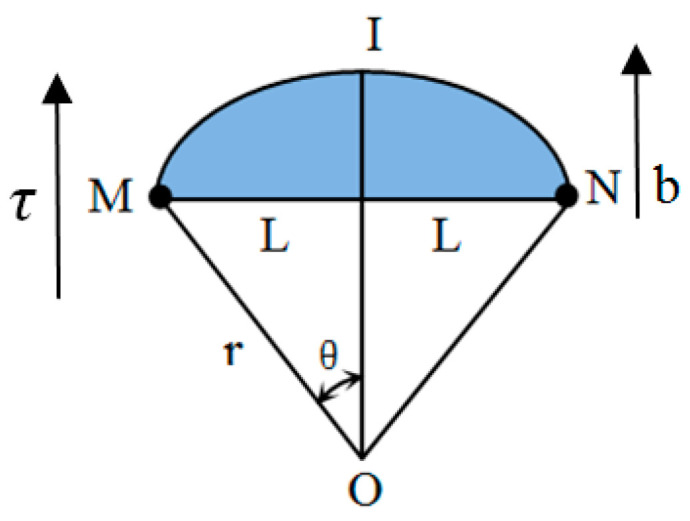
The dislocation chord model.

**Figure 2 materials-16-03363-f002:**
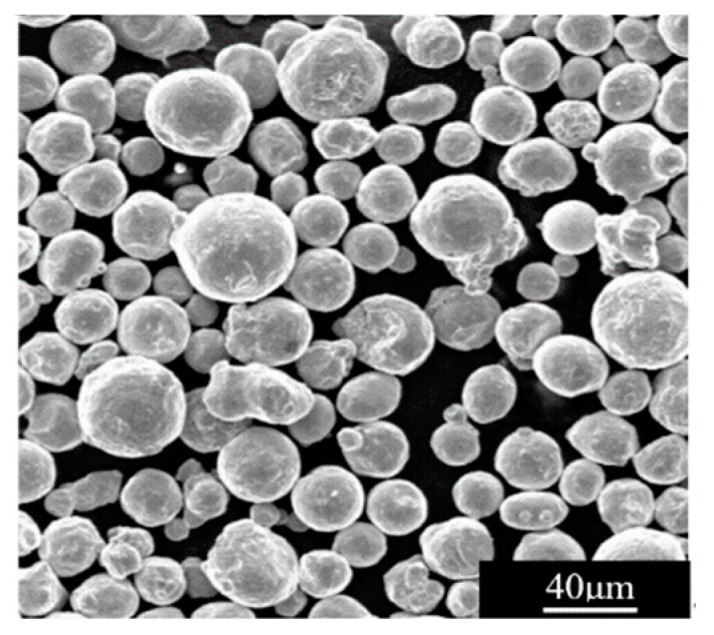
The SEM morphology of 316L stainless steel powder.

**Figure 3 materials-16-03363-f003:**
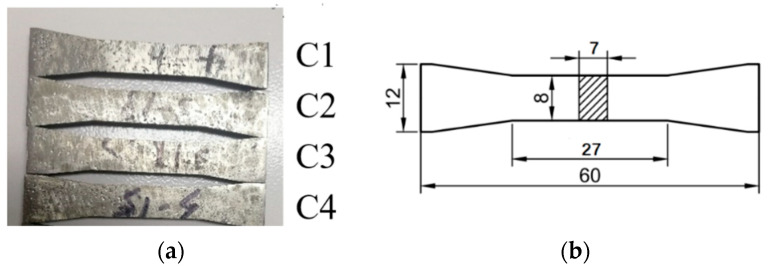
The SLM 316L stainless specimens: (**a**) the prepared specimens; (**b**) schematic diagram of the specimen (size in millimeters).

**Figure 4 materials-16-03363-f004:**
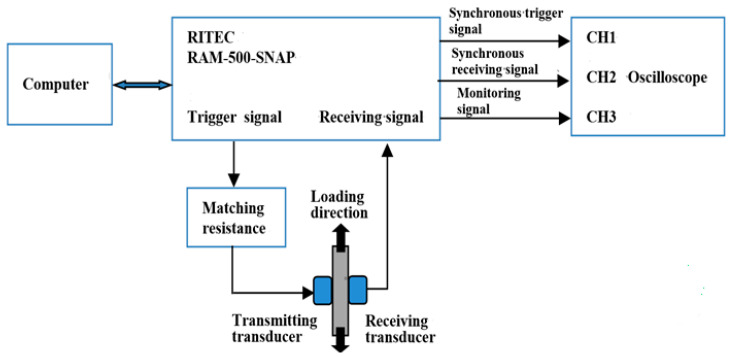
The experimental setup.

**Figure 5 materials-16-03363-f005:**
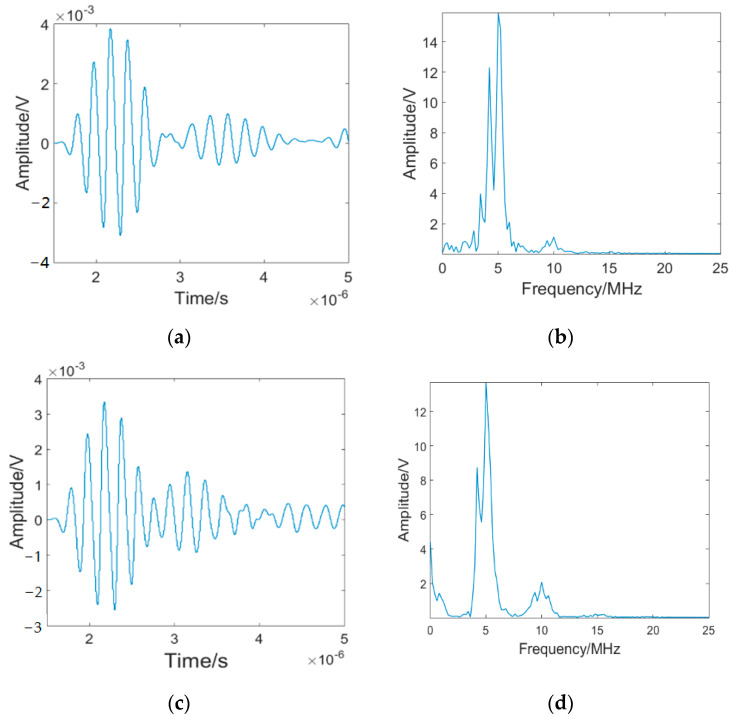
The ultrasonic testing signal for specimen C4 subjected to 2000 and 8000 fatigue cycles: (**a**) the received time-domain signal (2000 cycles); (**b**) Fourier spectra of the received signal (2000 cycles); (**c**) the received time-domain signal (8000 cycles); (**d**) Fourier spectra of the received signal (8000 cycles).

**Figure 6 materials-16-03363-f006:**
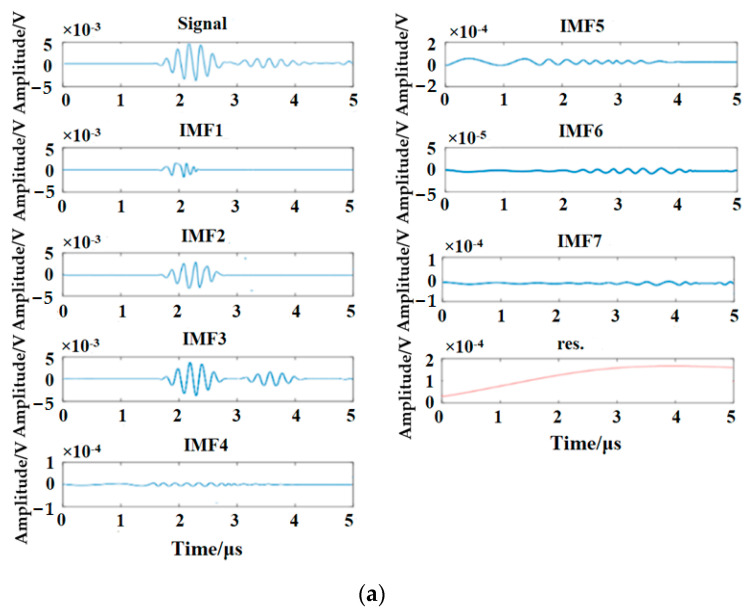
The results of EMD for the time-domain signals: (**a**) all IMFs of the ultrasonic testing signal (2000 fatigue cycles); (**b**) all IMFs of the ultrasonic testing signal (8000 fatigue cycles).

**Figure 7 materials-16-03363-f007:**
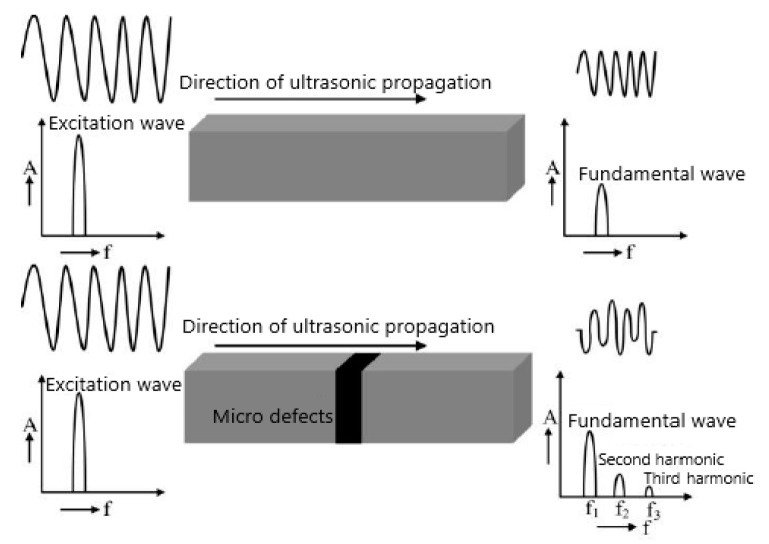
The principle of early damage detection based on higher harmonic testing.

**Figure 8 materials-16-03363-f008:**
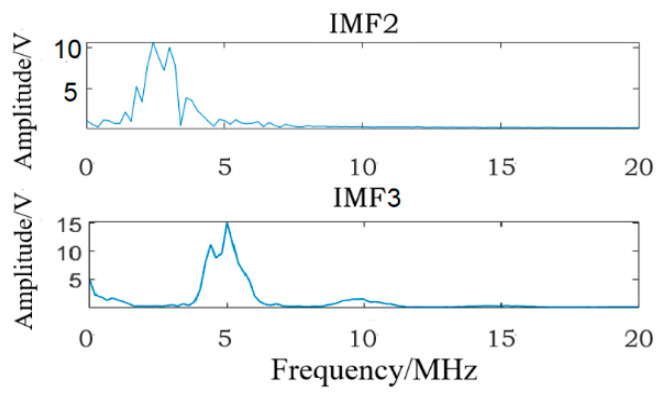
The frequency-domain analysis results of IMF2 to IMF3.

**Figure 9 materials-16-03363-f009:**
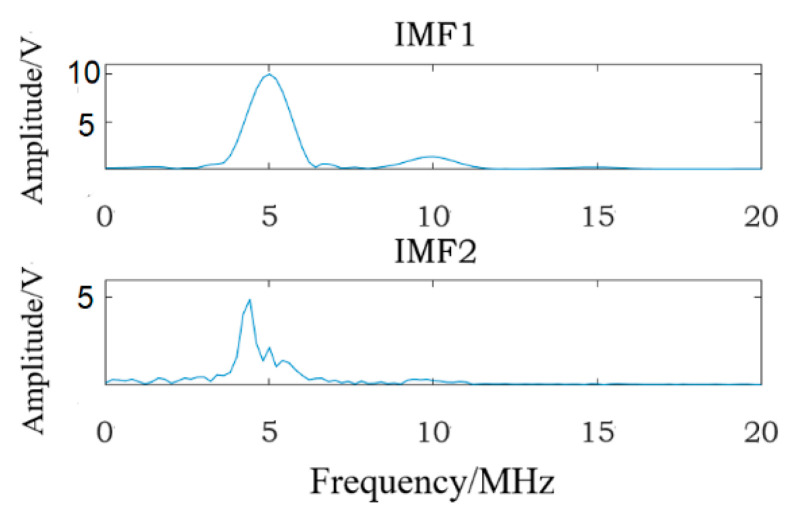
The frequency-domain analysis results of IMF1 to IMF2.

**Figure 10 materials-16-03363-f010:**
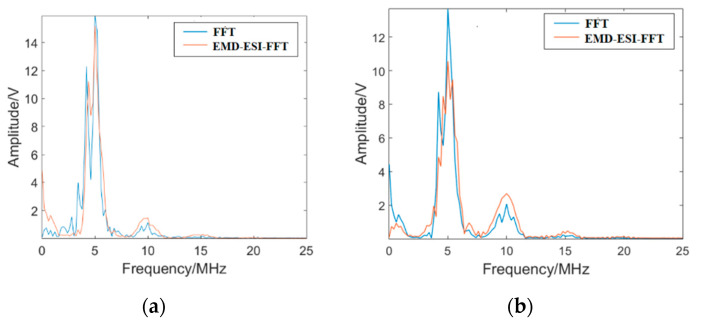
The comparison of FFT and EMD-ESI-FFT: (**a**) the ultrasonic signal (2000 fatigue cycles); (**b**) the ultrasonic signal (8000 fatigue cycles).

**Figure 11 materials-16-03363-f011:**
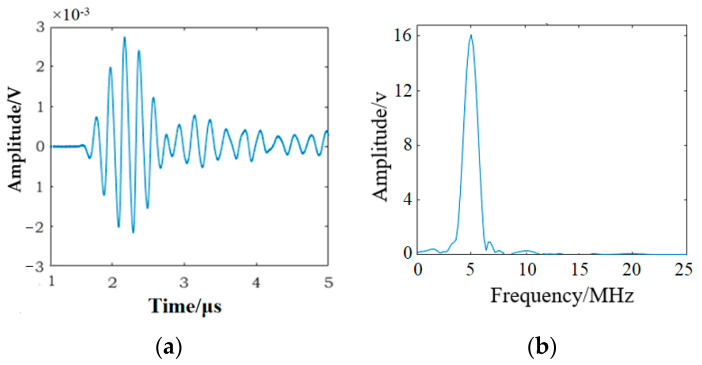
The ultrasonic signal for specimen C4 before fatigue test: (**a**) the time-domain signal; (**b**) EMD-ESI-FFT result of the time-domain signal.

**Figure 12 materials-16-03363-f012:**
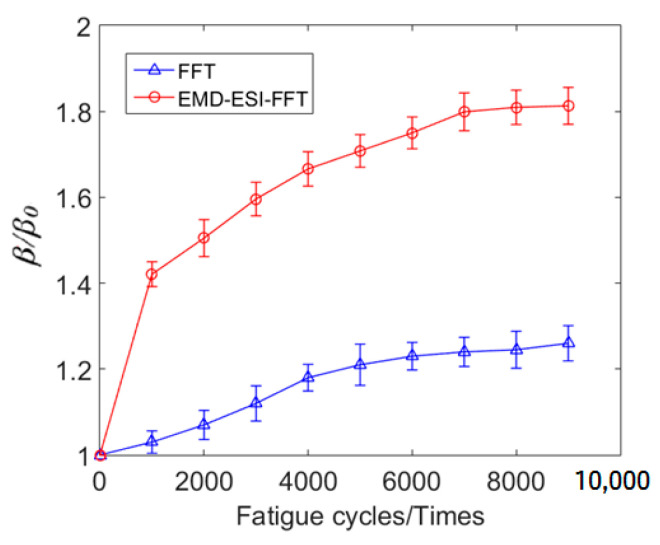
The relation curve between *β*/*β*_0_ and fatigue cycle number.

**Figure 13 materials-16-03363-f013:**
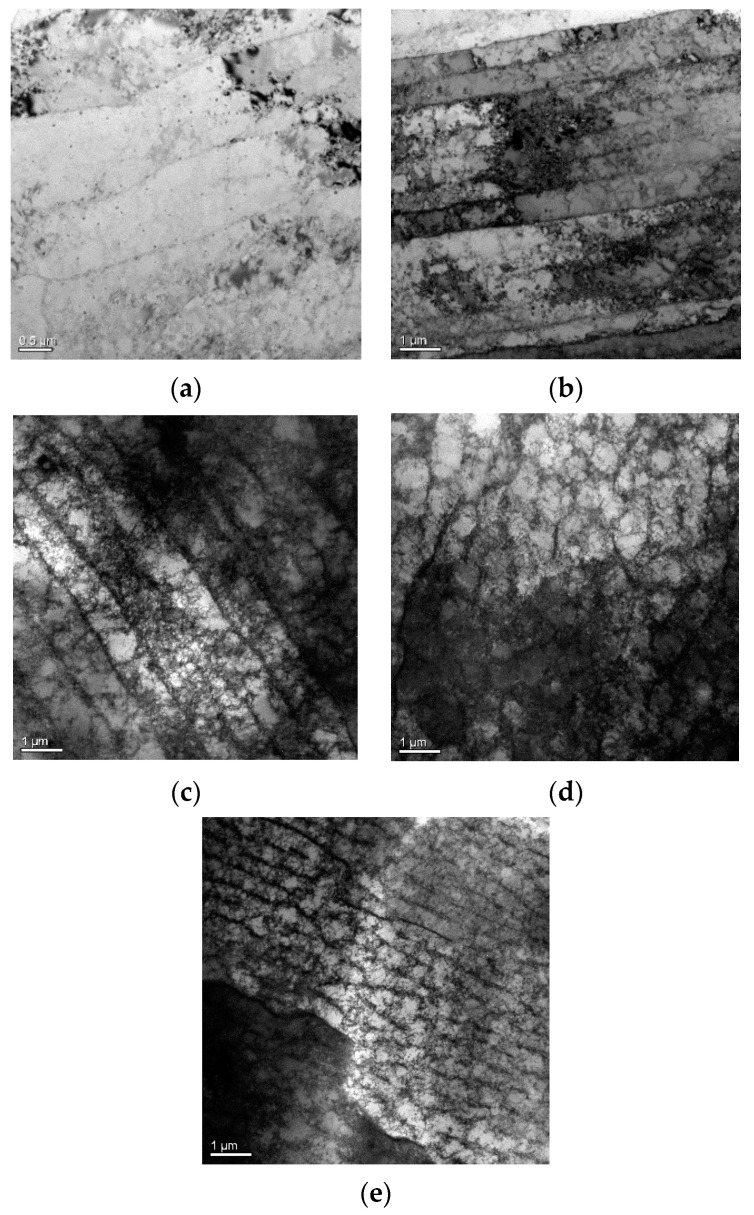
Microstructure photos of SLM 316L stainless steel fatigue specimen: (**a**) 1000 times, (**b**) 3000 times, (**c**) 5000 times, (**d**) 7000 times, (**e**) 9000 times.

**Table 1 materials-16-03363-t001:** Chemical composition of powdered 316L stainless steel.

Chemical Composition	Cr	C	Mo	Ni	Mn	Si	P	O	Fe
Mass Fraction (%)	17.6	0.04	2.05	12.05	0.3	0.85	0.04	<0.1	Bal.

**Table 2 materials-16-03363-t002:** Main fabrication parameters.

Scanning Speed (mm/s)	Laser Power (W)	Layer Thickness (μm)	Scanning Interval (μm)	Spot Diameter (μm)	Volume Fraction of Oxygen (%)
780	260	25	70	85	≤0.03

## Data Availability

The data presented in this study are available on request from the corresponding author.

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
