# Peer review of "A Novel Method for Early Fatigue Damage Diagnosis in 316L Stainless Steel Formed by Selective Laser Melting Technology"

_materials, 2023, doi:10.3390/ma16093363_

Round 1
Reviewer 1 Report
1. Abstract is very short and needs to be revised
2. objectives, experiments, important results, and an overall conclusion - include in abstract
3. Significant properties may be discussed briefly in the introduction
4. In the introduction part and enhance the core of this work with recently published articles
5. The end of the introduction should reflect this work's clear objective.
6. Improve the discussion section with some technical proof.
7. Language: Lot of grammatical errors, no space between words, and spelling errors. Kindly check this throughout the manuscript.
9. More results may be discussed in the conclusions.
10. Include the scope for future studies.
Language needs to be improved
Author Response
Dear Reviewer:
Thank you very much for the time and effort that you have put into reviewing the previous version of the manuscript. Your suggestions have enabled me to improve my work greatly. Appended to this letter is my point-by-point response to the comments raised by you. The comments are reproduced and my responses are given directly afterward in a different color (red).
Should you have any questions, please contact me without hesitate.
Kind regards
Yan XiaoLing

Reviewer 2 Report
The submission presents interested findings on identification of early fatigue damage diagnosis in 316L stainless steel using a novel approach. The submission needs following correction before its possible consideration for publication in the journal.
Methodology steps to be written in past tense (pg 4, row 132-139)
citation needs to be properly placed (pg 5, row 168)
Figure number needs correction (pg 6, row 182)
citation needs to be properly placed (pg 7, row 200)
Use better resolution for Figure 5 and 6
Reference 42 appears not being cited in the text, either remove it from the list of reference or cite at suitable place in the draft.
Minor correction on language is needed.
Methodology steps in some segments are written with present tense and needs to be corrected with past tense instead.
Author Response

(The authors gave the same response as above.)

Reviewer 3 Report
1. This paper studies early fatigue damage diagnosis in 316L stainless steel formed by selective laser melting technology using the EMD-ESI-FFT method. The paper provides useful information for further study and for practical applications in this field.
2. The paper is generally written well. Some grammatical errors and mistakes, though, need to be corrected. Some examples are listed below.
3. Strain "e" should use different number for equation 5 and 7; both named "1" is confusing.
4. Add unit in the title of Fig. 3.
5. Line 138, change "Computer" to computer. Lines 137-139, this reads not like a complete sentence, please reword it. Line 158, "316Lstainless specimen", add a space after L.
6. Lines 186-187, why repeat "the nonlinear coefficient"?
7. Some RITEC machine has "filter" function. Did the authors use such function to help reduce the noise at the higher hormonic level?
8. It is recommended for the authors to carefully proofread the paper before the next submission.
There are some grammatical errors that need to be corrected. Some sentences are too long.
Author Response

(The authors gave the same response as above.)
